# MULTI-EPOCH LEARNING WITH DATA AUGMENTATION FOR DEEP CLICK-THROUGH RATE PREDICTION

## ABSTRACT

This paper investigates the one-epoch overfitting phenomenon in Click-Through Rate (CTR) models, where performance notably declines at the start of the second epoch. Despite extensive research, the efficacy of MEL over the conventional one-epoch approach remains unclear. As a result, all potential rewards from MEL can hardly be obtained. We identify the overfitting of the embedding layer instead of the Multi-Layer Perceptron (MLP) layers, as the primary issue. To address this, we introduce a novel Multi-Epoch learning with Data Augmentation (MEDA) framework. We design algorithms for both non-incremental and incremental learning scenarios in the industry. MEDA minimizes overfitting by reducing the dependency of the embedding layer on trained data, and achieves data augmentation through training the MLP with varied embedding spaces. MEDA's effectiveness is established on our finding that pre-trained MLP layers can adapt to new embedding spaces and enhance model performances. This adaptability highlights the importance of the relative relationships among embeddings over their absolute positions. We conduct extensive experiments on several public and business datasets, and the effectiveness of data augmentation and superiority over conventional SEL are consistently demonstrated for both non-incremental and incremental learning scenarios. To our knowledge, MEDA represents the first universally reliable MEL strategy tailored for deep CTR prediction models. We provide theoretical analyses of the reason behind the effectiveness of MEDA. Finally, MEDA has exhibited significant benefits in a real-world incremental-learning online advertising system.

## 1 INTRODUCTION

Click-through rate (CTR) prediction is crucial in online recommendation and advertising systems, benefiting significantly from advancements in deep learning-based models (Cheng et al., 2016; Qu et al., 2016; Guo et al., 2017; Yu et al., 2020; Zhou et al., 2018; 2019; Pi et al., 2019; Li et al., 2022). Despite the progress and diverse approaches, including non-incremental learning for smaller datasets and incremental learning (Cai et al., 2022; Guan et al., 2022; Mi et al., 2020; Yang et al., 2023) for larger or real-time datasets, a common challenge persists: "one-epoch overfitting (OEO)" (Zhou et al., 2018; Zhang et al., 2022). This phenomenon, where model performance drops sharply at the beginning of the second training epoch, contrasts with other deep learning domains like computer vision (He et al., 2016; Russakovsky et al., 2015) and audio processing (Purwins et al., 2019), where multi-epoch learning (MEL) enhances model convergence. The OEO issue has been under investigation since 2018 (Zhou et al., 2018), yet a universally reliable solution remains elusive. As a consequence, it is hard for us to obtain *any potential benefits from MEL*. This includes the further convergence of models (especially for cold-start scenarios), the re-training necessary to rebuild MLP or mitigate catastrophic forgetting (Katsileros et al., 2022), and the implementation of "rethinking" training techniques (e.g., unsupervised domain adaptation (Wilson & Cook, 2020) or label-noise correction (Song et al., 2022)). Moreover, the OEO problem also affects large language models (LLM) (Ouyang et al., 2022; Komatsuzaki, 2019).

Current straightforward solutions (including regularization, dropout, and model simplification) for OEO can only mitigate the problem but can hardly solve it, except for simple datasets with very limited high-dimensional categorical features (Zhang et al., 2022; Zhou et al., 2018). And the only conclusion from existing research is that the OEO issue is related to feature sparsity (Zhang et al.,

Figure 1: Our proposed MEDA framework. For non-incremental learning, MEDA reinitializes the embedding parameters at the onset of each training epoch; for incremental learning, MEDA maintains multiple independently initialized embedding layers and for each dataset, trains each embedding layer once successively. The embedding layers can be selected based on requirements or costs.

2022). Therefore, to uncover the fundamental causes of OEO and to address it at its core, we introduce a novel Multi-Epoch learning with Data Augmentation (**MEDA**) framework, tailored for *both non-incremental and incremental learning scenarios in the industry*. Our framework can also cover both the classification and regression tasks. Specifically, *we identify the overfitting of the embedding layer instead of the MLP layers, caused by high-dimensional data sparsity, as the primary issue for OEO. Moreover, the embedding layer is overfitted even during the first epoch!* Then we design MEDA to effectively mitigate overfitting by decoupling the embedding layer and the data. In detail, in our non-incremental MEDA algorithm shown in Figure 1, the embedding-data dependency is reduced by reinitializing the embedding layer at the onset of each training epoch. *Note that, compared with Single-Epoch Learning (SEL), MEDA losses no information because both MEDA and SEL use the embedding layer trained once, and MEDA uses the MLP layer trained more epochs than SEL does.* The non-incremental MEDA is extended to the incremental MEDA to further reduce the additional embedding-MLP dependency in the incremental learning setting, which is shown in Figure 1. We leverage multiple independently initialized embedding layers—each for an extra epoch: for each dataset, each embedding layer can be selected to train once successively. The selection can be based on requirements such as computation/storage costs. Intuitively, on each dataset, each embedding layer in MEDA is trained once only, thereby minimizing overfitting, while the MLP layers are trained repeatedly to improve convergence. *Our proposed MEDA can be regarded as a data augmentation method because it can be treated as learning the MLP on the same categorical features with varied embedding spaces.* To our knowledge, MEDA represents the first universally reliable MEL strategy tailored for deep CTR prediction models.

We conduct comprehensive experiments on public and business datasets to show the effectiveness of data augmentation and superiority over SEL and straightforward MEL methods. Notably, MEDA's second-epoch performance consistently exceeds that of SEL across various datasets and CTR models, with improvements in test AUC ranging from 0.8% to 4.6%. This trend persists across multiple epochs without inducing overfitting, offering flexibility in training duration based on training-cost considerations. Our findings confirm that pre-trained MLP layers can adapt to new embedding spaces, enhancing performance without overfitting. *This adaptability underscores the MLP layers' role in learning a matching function focused on the relative relationships among embeddings rather than their absolute positions.* Furthermore, MEDA demonstrates remarkable efficiency by achieving or surpassing the outcomes of complete-data training of SEL with only a fraction of the data, e.g., in most cases, MEDA with $1/2$ data can outperform SEL with complete data, sometimes even 3 epochs on $1/8$ data can outperform 1 epoch on complete data, and *thus MEDA may boost performances in cold-start scenarios.* We provide theoretical analyses of the reason behind the effectiveness of MEDA in Appendix A.4. The successful deployment of MEDA in a live environment, corroborated by positive online A/B testing results, further attests to its practical value and impact.

## 2 RELATED WORKS

***One-Epoch Overfitting***. OEO has been studied since the work of Zhou et al. (2018). They have proposed a method named mini-batch aware regularization (MBA-reg) to approximate the $\ell_2$-regularization for computational efficiency to handle OEO. However, it only works on their simple dataset with very limited high-dimensional categorical features. The $\ell_2$-regularization has been found ineffective by subsequent empirical research of Zhang et al. (2022), and unstable and hard

to tune hyperparameters by our work. The research of Zhang et al. (2022) indicates that reducing feature sparsity can diminish the prevalence of OEO, yet the potential superiority of MEL over traditional SEL remains uncertain. Specifically, they have performed extensive experiments with straightforward approaches to reduce the sparsity, such as regularization, dropout, and model simplification (including ID hashing, ID filtering, reducing the embedding dimension, reducing the number of neurons or layers of the MLP), and changing batch sizes, activation functions, and optimization algorithms. They concluded that none of these methods can solve the OEO problem: these methods either still confront overfitting when the number of epochs is greater than one, or harm the capacity of the model such that the results are lower than direct SEL. They have concluded that OEO is related to feature sparsity, but cannot further uncover its fundamental causes. Parallel observations (Ouyang et al., 2022) in large language models undergoing supervised fine-tuning reveal a similar tendency towards OEO, albeit with a suggestion that a moderate level of overfitting might actually benefit downstream tasks. This concept of "appropriate overfitting" presents an intriguing avenue for future exploration within our proposed framework.

*Pre-training*. Recent approaches (Lin et al., 2023; Liu et al., 2022; Wang et al., 2023; Muhamed et al., 2021) have explored pre-training to enhance the representational capabilities of embedding and feature extraction layers within MLPs for various applications, yet these advancements fall short in demonstrating their efficacy in avoiding overfitting when CTR prediction is incorporated as an auxiliary training objective, nor do they facilitate MEL for such models. In contrast, graph learning research has delved into fine-tuning pre-trained models for new graphs, facing challenges related to either maintaining a consistent node ID space (Hu et al., 2019; Liu et al., 2023; Lu et al., 2021; Hu et al., 2020) or solely leveraging graph structure while neglecting node features (Qiu et al., 2020; Zhu et al., 2021). This leaves an open question in the context of CTR models: the potential for pre-trained MLP layers to contribute positively to a distinct embedding space remains unexplored and warrants further investigation.

## 3 BACKGROUND

CTR (Click-Through Rate) prediction models are distinguished by their handling of high-dimensional sparse data, often involving billions of categorical features, e.g., User ID, Item ID, and user behaviors (lists of watched/clicked Item IDs of each user), with low occurrence rates (Jiang et al., 2019; Zhao et al., 2019; 2020). To handle these categorical features, deep CTR prediction models typically adopt an embedding layer (Zhang et al., 2016) at the front, followed by various types of MLP structures, with the embedding layer responsible for mapping the high-dimensional categorical features to low-dimensional vectors. Given the concatenated dense representation vector, an MLP is employed to capture the nonlinear interaction among features (Liu et al., 2020).

## 4 METHODOLOGY

In this section, we present the detailed methodology of our method. First, we define the notations and settings of our study. Then we will introduce our problem identification and proposed frameworks.

***Non-incremental Learning***. Consider a dataset $\mathcal{D} = \{(\mathbf{x}^i, \mathbf{y}_i)\}_{i=1}^n$ consisting of $n$ independent samples. For the $i$th sample, $\mathbf{x}^i \in \mathcal{X} \subset \mathbb{R}^d$ is a feature vector with $d$ dimensions, $y^i \in \mathcal{Y} \subset \mathbb{R}^m$ is the label of the $i$th sample. Note that this setting covers both classification and regression tasks. Let $M \in \mathcal{M}$ be a data-driven model. Specifically, let $\boldsymbol{\theta}$ be the collection of training parameters of the MLP layers, and $\mathbf{E}$ be the collection of training parameters of the embedding layer. Let $A \in \mathcal{A}$ be a training algorithm, and we denote by $M = A(\{\mathcal{D} : k\})$ as obtaining the model $M$ by training the dataset $\mathcal{D}$ by algorithm $A$ for $k \in \mathbb{Z}_+$ epochs. And we denote by $S(\mathcal{D} \mid M) \in \mathbb{R}$ as the evaluation score (the bigger the better) obtained from evaluating $M$ on $\mathcal{D}$. Finally, splitting $\mathcal{D}$ into $\mathcal{D}_{tr}, \mathcal{D}_{te}$ as the training and testing datasets, respectively, our goal is to see if there exists a $k > 1$ such that $S(\mathcal{D}_{te} \mid A(\{\mathcal{D}_{tr} : k\})) > S(\mathcal{D}_{te} \mid A(\{\mathcal{D}_{tr} : 1\}))$.

***Incremental Learning***. Consider the training dataset $\mathcal{D}_{tr} = \{\mathcal{D}_{tr}^t\}_{t=1}^T$ consisting of $T$ successive sub-datasets. Our goal is to see if there exists a $k^t > 1$ such that $S(\mathcal{D}_{te} \mid A(\{\mathcal{D}_{tr}^t : k^t\}_{t=1}^T)) > S(\mathcal{D}_{te} \mid A(\{\mathcal{D}_{tr}^t : 1\}_{t=1}^T))$, where we denote by $A(\{\mathcal{D}_{tr}^t : k^t\}_{t=1}^T)$ as training each $\mathcal{D}_{tr}^t$ for $k^t$ epochs and denote by $A(\{\mathcal{D}_{tr}^t : 1\}_{t=1}^T)$ as training each $\mathcal{D}_{tr}^t$ for 1 epoch.

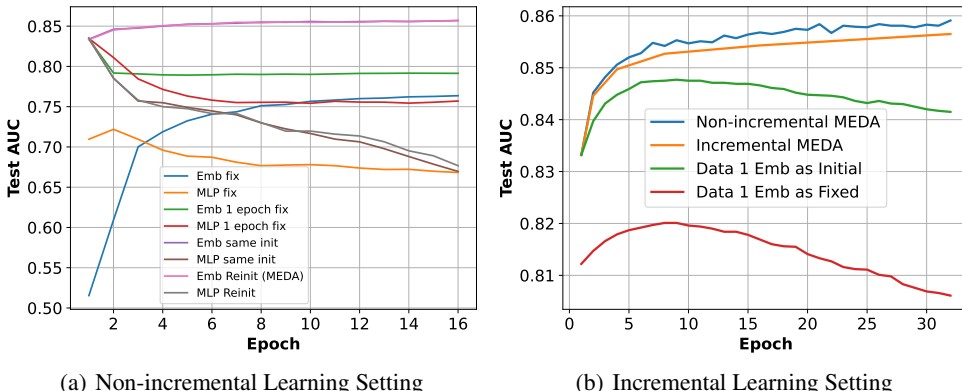

(a) Non-incremental Learning Setting  (b) Incremental Learning Setting

Figure 2: (a) The test AUC curves for training DNN on the Amazon dataset with different training paradigms. (b) The test AUC curves of DNN on the Amazon dataset, comparing different variants of incremental MEDA to train $\mathcal{D}_{tr}^2$ multiple times. The incremental MEDA has run 2, 4, 8, 16, and 32 epochs. Note that the results of MEDA methods are only for reference because they also train $\mathcal{D}_{tr}^1$ multiple times.

## 4.1 PROBLEM IDENTIFICATION

In Figure 2, we show that *the primary causative factor of OEO is overfitting of the embedding instead of the MLP*. We treat the embedding and MLP as two factors and design control strategies to compare their effects. First, we test the "Emb (MLP) fix" strategy: fixing the embedding (MLP) and training the MLP (embedding). Figure 2 (a) shows that the MLP does not overfit when the embedding is fixed, while the embedding overfits when the MLP is fixed. We further test "Emb (MLP) 1 epoch fix": after 1 epoch of joint training of the embedding and MLP, fixing the embedding (MLP) and training the MLP (embedding). The same phenomenon happens after the second epoch, although overfitting occurs at the second epoch. Then we test two reinitialization strategies: "Emb (MLP) same init: for each epoch, using the same initialization result to reinitialize the embedding (MLP)" and "Emb (MLP) Reinit: for each epoch, independently reinitialize the embedding (MLP)". It shows that, when the embedding is reinitialized, for the first time, the OEO is solved: the test AUC steadily improves across epochs. While reinitializing the MLP cannot stop the overfitting. These results show that only when the embedding is controlled, the overfitting can be controlled or avoided altogether, and thus reveal that OEO primarily stems from embedding (instead of MLP) overfitting. This discrepancy is likely due to the sparse nature of high-dimensional data, where a vast number of categorical values exist but each appears infrequently. Consequently, embedding vectors, representing these infrequent values, are prone to overfitting due to limited training samples. Meanwhile, MLP parameters, engaging with the entire dataset, exhibit a lower risk of overfitting.

Moreover, in Figure 2 (b), we show that *the embedding overfits even during the first epoch.* We divid the training data $\mathcal{D}_{tr}$ into two parts based on time: $\mathcal{D}_{tr}^1$ and $\mathcal{D}_{tr}^2$. Then we test two strategies: "Data 1 Emb as Initial: using the final embedding of training $\mathcal{D}_{tr}^1$ as the *learnable initial* embedding for MEL of $\mathcal{D}_{tr}^2$." and "Data 1 Emb as Fixed: using the final embedding of training $\mathcal{D}_{tr}^1$ as the *fixed* embedding for MEL of $\mathcal{D}_{tr}^2$.". Both strategies exhibit overfitting beyond epoch 10. Comparing between "Data 1 Emb as Fixed" and "Emb fix" in Figure 2 (a), a trained embedding causes overfitting while an untrained embedding does not, which can also be concluded from comparing between "Data 1 Emb as Initial" and "Emb same init". Note that, both trained embeddings are not trained on $\mathcal{D}_{tr}^2$ which is for MEL. Therefore, we show that the embeddings trained only once on $\mathcal{D}_{tr}^1$ are already problematic before being trained for the second epoch. Thus, the MLPs trained with multi-epochs on the problematic embeddings cause overfittings.

Then we ask, *which data samples does the embedding overfit on?* Figure 3 shows that *the embedding overfits on each trained data sample*, comparing the loss curves without MEDA in training and testing, since the overfitting emerges exactly at the beginning of the second training epoch, indicating the initial embedding of the second epoch precisely memorizes the information of any data sample in the first epoch.

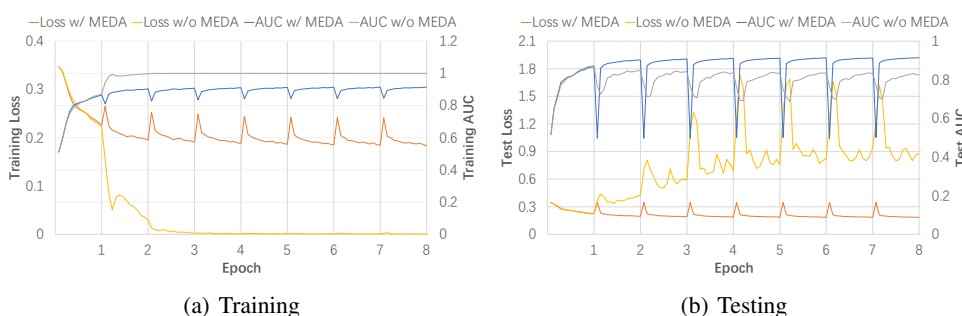

(a) Training            (b) Testing

Figure 3: The training/testing metric curves of training DNN on the Taobao dataset, with or without non-incremental MEDA.

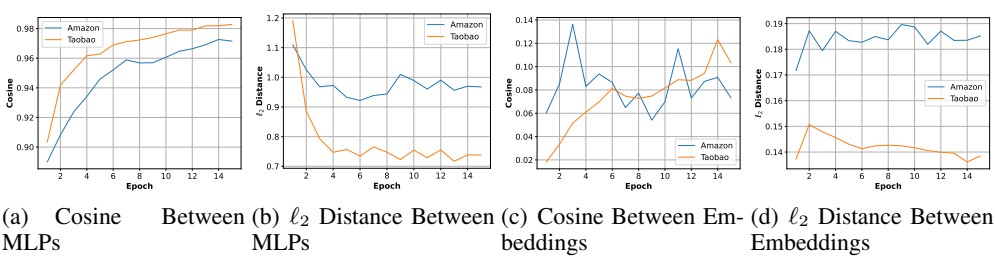

(a) Cosine Between MLPs    (b) $\ell_2$ Distance Between MLPs    (c) Cosine Between Embeddings    (d) $\ell_2$ Distance Between Embeddings

Figure 4: The parameter-convergence metric curves of non-incremental MEDA using DNN on the public datasets. Each panel shows metrics between parameters of two successive epochs varying across epochs.

Finally, *what information does the embedding overfit in each sample?* Our findings suggest that, *the embedding overfits the absolute positions of embeddings*. Because, first, Figure 2 (a) shows that the embedding reinitialization strategies do not harm the performances across epochs. Second, Figure 4 (c) and (d) show that the similarities of absolute positions between two successive final embeddings are low across epochs. Therefore, the absolute positions of samples are not important but may be over-learned by the embedding.

## 4.2 OUR MEDA FRAMEWORK

As introduced, for MEL, we propose MEDA to avoid the OEO.

***Non-incremental Problem Formulation***. Based on the findings discussed, the initial embedding of each epoch is crucial for mitigating OEO and must be devoid of any exact information from trained samples. A straightforward solution is to randomize the initial parameters, ensuring their independence from trained data. Therefore, we propose the novel strategy of randomly initializing embedding parameters at the start of each epoch in our non-incremental MEDA framework. See Algorithm 1 for details.

One might think that reinitializing embedding parameters may cause information loss. *While we should note that, compared with SEL, MEDA losses no information because both MEDA and SEL use the embedding layer trained once, and MEDA uses the MLP layer trained more epochs than SEL does.* Therefore, we can regard the multi-epoch of training as just pre-training the MLP for a final regular SEL. And we indeed show that such pre-training is effective: the performances improve steadily across epochs. The essential insight is that for CTR model MLP layers, *the precise values or absolute positions of embeddings are less critical than their interrelations*. This understanding allows us to view the additional data samples with different embeddings while maintaining crucial semantic relationships as *augmented data samples*. Furthermore, our results in Figure 4 (a) demonstrate that the MLP is indeed nearing convergence throughout the MEL.

---

**Algorithm 1** Non-incremental MEDA

---

**Input:** Training dataset $\mathcal{D}_{tr}$, training algorithm $A$, the number of training epoch $k$.
**Output:** MLP parameters $\hat{\theta}$ and embedding parameters $\hat{\mathbf{E}}$.
1: Random initialize $\tilde{\theta}_0$.
2: **for** epoch $r = 1$ to $k$ **do**
3:     Initialization: Random initialize $\mathbf{E}_r$. $\theta_r = \tilde{\theta}_{r-1}$.
4:     Training and Update: $\tilde{\theta}_r, \tilde{\mathbf{E}}_r = A(\{\mathcal{D}_{tr} : 1\})$ with $\theta_r, \mathbf{E}_r$ as the initial parameters.
5: **end for**
6: **return** $\hat{\theta} = \tilde{\theta}_k, \hat{\mathbf{E}} = \tilde{\mathbf{E}}_k$.

---

---

**Algorithm 2** Incremental MEDA

---

**Input:** Training dataset $\mathcal{D}_{tr} = \{\mathcal{D}_{tr}^t\}_{t=1}^T$, training algorithm $A$, the max number of training epoch $k$.
**Output:** MLP parameters $\hat{\theta}$ and embedding parameters $\hat{\mathbf{E}}$.
1: Random initialize $\theta^c, \{\mathbf{E}_r^0\}_{r=1}^k$.
2: **for** dataset index $t = 1$ to $T$ **do**
3:     **for** epoch $r = 1$ to $k$ **do**
4:       **if** $\mathbf{E}_r^{t-1}$ is selected based on requirements such as computation/storage costs **then**
5:         Initialization: $\theta_r = \theta^c$.
6:         Training and Update: $\tilde{\theta}_r, \mathbf{E}_r^t = A(\{\mathcal{D}_{tr}^t : 1\})$ with $\theta_r, \mathbf{E}_r^{t-1}$ as the initial parameters.
7:         $\theta^c = \tilde{\theta}_r, \mathbf{E}^c = \mathbf{E}_r^t$.
8:       **else**
9:         $\mathbf{E}_r^t = \mathbf{E}_r^{t-1}$.
10:       **end if**
11:     **end for**
12: **end for**
13: **return** $\hat{\theta} = \theta^c, \hat{\mathbf{E}} = \mathbf{E}^c$.

---

***Incremental Problem Formulation***. In an incremental learning framework, where datasets are processed successively, we encounter a unique challenge: OEO also occurs upon the second training of the $t$th dataset $t > 1$, involving both embedding and MLP layer optimization. This scenario diverges from the non-incremental setting, as reinitializing embedding parameters at the start of the $t$th dataset's training will disregard the accumulated knowledge from datasets 1 to $(t-1)$, which is undesirable. Based on our findings, to prevent OEO, the initial embedding parameters should not contain *exact* information in any data sample in the $t$th dataset, but should contain information in datasets $1 \sim (t-1)$. Therefore, one option involves adopting the final embedding parameters from the $t-1$th dataset's training $\mathbf{E}^{t-1}$. Nonetheless, our findings in Section 4.1 show that $\mathbf{E}^{t-1}$ is already problematic because it overfits absolute positions of embeddings. Thus, drawing from insights in the non-incremental MEDA, we propose to leverage multiple $\mathbf{E}^{t-1}$s with different positions to perform data augmentation. Specifically, we independently initialize multiple groups of embedding parameters to form distinct embedding spaces. These are then trained sequentially with the MLP on each dataset. See Algorithm 2 for details. Our non-incremental MEDA can benefit more from data augmentation than our incremental MEDA because the differences between final embeddings are larger than those in our incremental MEDA due to training on more data in an epoch, while our incremental MEDA can benefit more from embedding consistency.

### 4.3 COMPUTATION/STORAGE COMPLEXITY ANALYSES

Since our method only adds initialization processes, which are negligible for the computation complexity of training. Therefore, our method adds negligible computation complexity compared with standard MEL. On the other hand, for the incremental learning setting, our method requires $\mathcal{O}(kND)$ storage resources to maintain $k$ groups of embedding parameters for $k$-epoch learning, with $N$ representing the number of IDs and $D$ embedding vector dimension.

## 5 EXPERIMENTS

In this section, we present the experimental setup and conduct extensive experiments to evaluate the effectiveness and superiority of our proposed MEDA framework, along with online A/B test results. Ablation studies of hyperparameter robustness are in Appendix A.3 due to limited space.

### 5.1 EXPERIMENTAL SETUP

***Datasets***. We conduct comprehensive evaluations on two public datasets and two business datasets.

**Amazon dataset**[1]. It is a frequently used *public* benchmark that consists of product reviews and metadata collected from Amazon (Ni et al., 2019), with 51 million records, 1.5 million users, 2.9 million items, and 1252 categories. In our study, we adopt the Books category of the Amazon dataset. We predict whether a user will review an item.

**Taobao dataset**[2]. It is a *public* compilation of user behaviors for CTR prediction from Taobao's recommender system (Zhu et al., 2018), with 89 million records, 1 million users, 4 million items, and 9407 categories.

**Short-Video Order (SVO) dataset**. It is our collected *large business* dataset that consists of user behaviors from a large video recommender system, with 1.75 billion records, 0.2 billion users, 6 million items, and 1259 categories. For this dataset, we predict the order behaviors of each user. We split 6 days for training and 1 day for testing.

**Short-Video Search LTV (SVSL) dataset**. It is also our collected *business* dataset that consists of user behaviors from a large video *search* system, with 0.3 billion records, 40 million users, 0.5 million items, and 324 categories. For this dataset, we predict the Life-Time Value (LTV) (Theocharous et al., 2015) value of each order for each user. We split 370 days for training and 1 day for testing.

For incremental learning, for both the public and the SVO datasets, we split the first half of training data as $\mathcal{D}_{tr}^1$ and the rest as $\mathcal{D}_{tr}^2$, while for the SVSL dataset, we split the first 280 days of training data as $\mathcal{D}_{tr}^1$ and the rest as $\mathcal{D}_{tr}^2$.

***CTR Models and Metrics for Evaluation***. We apply our method on the following CTR Models: **DNN** is a base deep CTR model, consisting of an embedding layer and a feed-forward network with ReLU activation. **DIN** (Zhou et al., 2018) proposes an attention mechanism to represent the user interests w.r.t. candidates. **DIEN** (Zhou et al., 2019) uses GRU to model user interest evolution. **MIMN** (Pi et al., 2019) proposes a memory network-based model to capture multiple channels of user interest drifting for long-term user behavior modeling. **ADFM** (Li et al., 2022) proposes an adversarial filtering model on long-term user behavior sequences. For the business datasets, we adopt DIN as default. We denote our non-incremental and incremental MEDA as **MEDA-NI** and **MEDA-I**, respectively. Our methods equal SEL when the number of epoch is 1. For binary classification tasks, i.e., click or order prediction, we use Area under the curve (AUC) and binary cross-entropy loss as evaluation metrics, while for the regression tasks, i.e., the LTV prediction, we use AUC score between the LTV prediction scores and the binary order labels as the evaluation metric, following the common business practice.

***Implementation Details***. All CTR Models adhere to the optimal hyperparameters reported in their respective papers. For public datasets, we adopt Adam (Kingma & Ba, 2014) as the optimizer with a learning rate of 0.001, and Glorot (Glorot & Bengio, 2010) as the initializer for embedding parameters. For business datasets, we adopt Adagrad (Duchi et al., 2011) as the optimizer with a learning rate of 0.01, and uniform initializer with the range of 0.01. Other details are in Appendix A.1.

### 5.2 EFFECTIVENESS AND SUPERIORITY EVALUATION

***Problem Justification***. In Figure 5 highlights the presence and substantial impact of OEO. In both the Amazon and Taobao datasets, the test AUC rapidly declines starting from the second epoch of the direct MEL. Whereas our MEDA can effectively improve the test AUC with the increase of

---

[1] https://nijianmo.github.io/amazon/index.html
[2] https://tianchi.aliyun.com/dataset/649

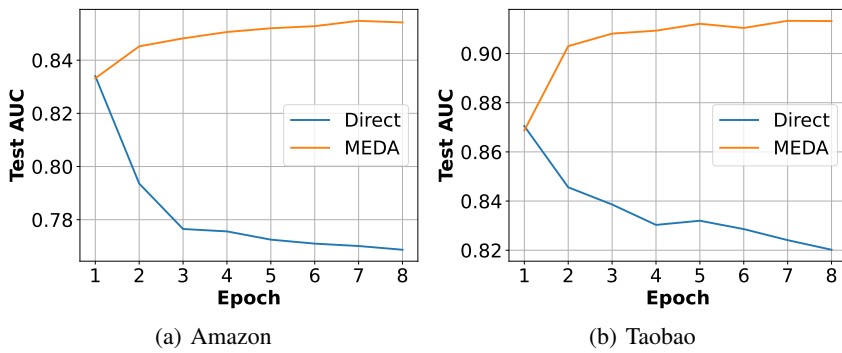

Figure 5: The test AUC curves of the Direct MEL and our non-incremental MEDA on the public datasets.

Table 1: The test AUC performance on the public datasets. MEDA methods run 2 epochs.

(a) Amazon

|  | DNN | DIN | DIEN | MIMN | ADFM |
|---|---|---|---|---|---|
| Single-Epoch | 0.8355 | 0.8477 | 0.8529 | 0.8686 | 0.8428 |
| MEDA-NI | **0.8450** | **0.8617** | **0.8602** | **0.8861** | **0.8507** |
| Improv. | +0.95% | +1.4% | +0.73% | +1.75% | +0.79% |
| MEDA-I | **0.8446** | **0.8588** | **0.8587** | **0.8832** | **0.8516** |
| Improv. | +0.91% | +1.11% | +0.58% | +1.46% | +0.88% |

(b) Taobao

|  | DNN | DIN | DIEN | MIMN | ADFM |
|---|---|---|---|---|---|
| Single-Epoch | 0.8714 | 0.8804 | 0.9032 | 0.9392 | 0.9462 |
| MEDA-NI | **0.9034** | **0.9265** | **0.9262** | **0.9500** | **0.9568** |
| Improv. | +3.2% | +4.61% | +2.3% | +1.08% | +1.06% |
| MEDA-I | **0.9054** | **0.9321** | **0.9281** | **0.9565** | **0.9549** |
| Improv. | +3.40% | +5.17% | +2.49% | +1.73% | +0.87% |

Table 2: The test AUC performance on the business datasets. MEDA methods run 2 epochs.

|  | Short-Video Order | Short-Video Search LTV |
|---|---|---|
| Single-Epoch | 0.8489 | 0.8184 |
| MEDA-NI | **0.8522** | **0.8248** |
| Improv. | +0.33% | +0.64% |
| MEDA-I | **0.8513** | **0.8233** |
| Improv. | +0.24% | +0.49% |

epoch without overfitting. The overfitting issue is more pronounced in the Amazon dataset due to its higher data sparsity (less data, more IDs).

***Evaluation of Superiority Over SEL***. Tables 1 and 2 highlight the significant superiority of our MEDA approach over conventional SEL. The corresponding results of test losses are in Appendix A.2. Our non-incremental and incremental MEDA methods outperform SEL on both public and business datasets by a substantial margin, which aligns with the improvement magnitude of each CTR model. Furthermore, incremental MEDA slightly outperforms non-incremental MEDA on the Taobao dataset but slightly underperforms on the Amazon, SVO, and SVSL datasets. This suggests that data augmentation in incremental MEDA is weaker, and data sparsity is more severe in these three datasets. The results presented for 2 epochs of MEDA are reasonable for most industrial applications, considering computation and storage costs. Additionally, Figure 6 demonstrates stable increases in test AUCs for most models as the number of epochs increases. Hence, it is feasible to determine the stopping point at any epoch, as the AUC does not significantly decrease after a certain number of epochs. This user-friendly feature enables users to select the number of epochs based on training costs.

***Evaluation of Superiority Over MEL***. We conduct experiments for straightforward MEL methods, including ID hashing (with DNN), batch normalization (with DNN), and $\ell_2$-regularization/MBA-reg (Zhou et al., 2018) (with DIN), the conclusion is the same as that of Zhang et al. (2022): these methods either still confront overfitting when the number of epochs is greater than one, or harm the capacity of the model such that the results are lower than directly single-epoch learning. The results are shown in Figure 7. ID hashing itself compromises the accuracy due to hash collisions. Large hash sizes (50k and 500k) still confront overfitting when the number of epochs is greater than one, while a small hash size of 5k harms the capacity of the model such that the results are lower than directly single-epoch learning. BN can improve the results slightly but still confronts overfitting when the number of epochs is greater than one. $\ell_2$-regularization/MBA-reg either still

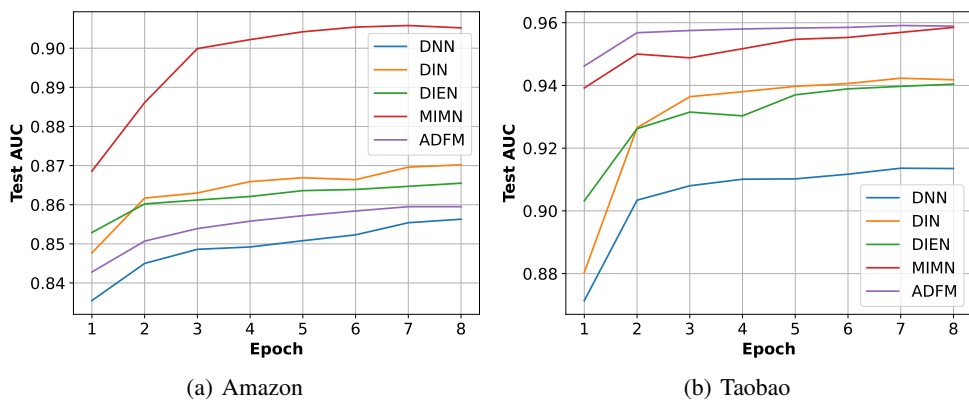

(a) Amazon

(b) Taobao

Figure 6: The test AUC curves of various models trained with our non-incremental MEDA on the public datasets.

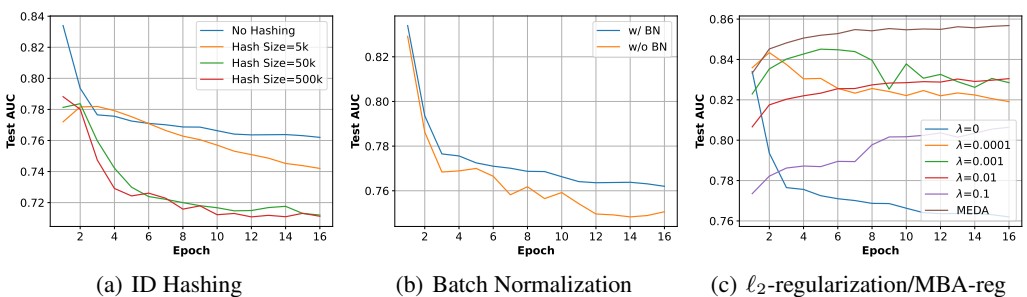

(a) ID Hashing

(b) Batch Normalization

(c) $\ell_2$-regularization/MBA-reg

Figure 7: The test AUC curves of the ID hashing, batch normalization, and $\ell_2$-regularization/MBA-reg on the Amazon dataset. The $\lambda$ in the figure denotes the regularization coefficient.

confront overfitting (e.g, $\lambda = 0.0001, 0.001$), or harm the capacity of the model such that the results are lower than direct single-epoch learning (e.g, $\lambda = 0.01, 0.1$). And our method non-incremental MEDA outperforms for each $\lambda$ and each epoch. Here we speculate on why MBA-reg can succeed in the DIN paper of Zhang et al. (2022). The success of MBA-reg is only reported on the Alibaba dataset which is their business dataset and not published. The success may be due to the specific properties of the dataset. For example, the high dimensional sparsity problem may be much less severe on their dataset. Since the high dimensional sparsity problem is the core problem of the OEO problem, then the overfitting may be also less severe on their dataset. As shown in Figure 7 (c), if the regularization coefficient is large (e.g., 0.01), the OEO indeed will not occur due to constrained model capacity. However, the overall results across epochs will be much worse than direct SEL. Since the test AUC results reported on the Alibaba dataset are relatively low (only around or even below 0.6), the success of MBA-reg may be due to the model capacity being constrained too low.

***Effectiveness of Data Augmentation***. Figure 3 illustrates the behavior of MEDA during training, where the use of MEDA results in a gradual decrease in training loss from the second epoch onwards, similar to encountering new data. Furthermore, Table 3 demonstrates that MEDA achieves comparable test AUC to SEL with fewer data and thus can boost performances in cold-start scenarios. In most cases, MEDA with half the data surpasses SEL with complete data, especially on Taobao, which validates the efficacy of data augmentation. The relatively better performance on Taobao compared to Amazon suggests that Amazon exhibits more severe data sparsity, resulting in weaker performance when joining multiple samples. Notably, in the case of ADFM on Taobao, even 3 epochs with $1/8$ of the data outperform a single epoch with complete data. This may be attributed to ADFM's extensive behavior window and increased interactions between ID features, as MEDA enhances the importance of ID relationships, providing more opportunities for improvements.

Table 3: The numbers of training epochs required for non-incremental MEDA on the Taobao/Amazon dataset with different data keeping rates of $\rho$s to achieve the test AUC with one-epoch on the complete data and the corresponding test AUCs.

(a) Taobao

| $\rho$ | | DNN | DIN | DIEN | MIMN | ADFM |
|---|---|---|---|---|---|---|
| 100% | #Epochs | 1 | 1 | 1 | 1 | 1 |
| | Test AUC | 0.8714 | 0.8804 | 0.9032 | 0.9392 | 0.9462 |
| 50% | #Epochs | **2** | **2** | **3** | **3** | **2** |
| | Test AUC | 0.8864 | 0.8989 | 0.9139 | 0.9444 | 0.9525 |
| 25% | #Epochs | **4** | **3** | **6** | **13** | **2** |
| | Test AUC | 0.8802 | 0.8847 | 0.9048 | 0.9395 | 0.9466 |
| 12.5% | #Epochs | **7** | **7** | **16** | 16* | **3** |
| | Test AUC | 0.8716 | 0.8844 | 0.9030 | 0.9287 | 0.9470 |

(b) Amazon

| $\rho$ | | DNN | DIN | DIEN | MIMN | ADFM |
|---|---|---|---|---|---|---|
| 100% | #Epochs | 1 | 1 | 1 | 1 | 1 |
| | Test AUC | 0.8355 | 0.8477 | 0.8529 | 0.8686 | 0.8428 |
| 50% | #Epochs | **10** | **4** | 16* | **4** | **7** |
| | Test AUC | 0.8370 | 0.8551 | 0.8481 | 0.8879 | 0.8441 |
| 25% | #Epochs | 16* | 16* | 16* | 16* | 16* |
| | Test AUC | 0.8268 | 0.8446 | 0.8337 | 0.8578 | 0.8319 |
| 12.5% | #Epochs | 16* | 16* | 16* | 16* | 16* |
| | Test AUC | 0.8093 | 0.8262 | 0.8157 | 0.8328 | 0.8202 |

\* means with the epoch number MEDA does not outperform SEL on the complete data.

Table 4: Online A/B Test Performance.

| | Test AUC | Retention | Revenue | Expected Revenue |
|---|---|---|---|---|
| Improv. | +0.14% | +6.6% | +0.32% | +0.91% |

***Effectiveness of MLP Convergence***. In Figure 4 (a) and (b), our MEDA approach is shown to enhance MLP convergence, as evidenced by the increasing similarity between two sets of MLP parameters in successive epochs. The cosine similarity between parameter groups continues to rise with each epoch, while the $\ell_2$ distance ceases to decrease after epoch 6, indicating that parameter direction is more crucial for CTR models than parameter distance. Moreover, the substantial discrepancy in final embedding parameters even at epoch 15 underscores the data augmentation effect of MEDA. Interestingly, the embedding parameters exhibit convergence on Taobao based on cosine similarity but not notably on Amazon, possibly due to the challenges posed by severe data sparsity in learning similar embedding patterns across epochs.

## 5.3 ONLINE RESULTS

We conduct an online A/B test on a large industrial video advertising platform, focusing on retention prediction. This platform processes billions of user requests daily, with millions of item candidates. It incorporates a four-stage recommender system: candidate retrieval, pre-ranking, ranking, and reranking, each progressively narrowing down the item selection for users. We deploy MEDA in the ranking module. The baseline method performs SEL, while we adopt our incremental MEDA to conduct the online A/B experiment. The experiment spanned 9 days, with 10% of the total online traffic allocated for both the baseline and our MEDA approach. Results in Table 4 demonstrate that MEDA significantly enhances the test AUC, user retention, and *overall* platform rewards (as evaluated by revenue and revenue expected by clients). This marks the first universally reliable solution addressing overfitting in MEL of large-scale sparse models for advertising recommendations. In our scenario, achieving satisfactory model performance typically necessitates training on one month's worth of data. Remarkably, with MEDA, comparable results can be obtained after just two weeks of training. Thus, implementing MEDA substantially reduces the required sample size and training costs while maintaining equivalent outcomes.

## 6 CONCLUSION

In this paper, we propose a novel Multi-Epoch learning with Data Augmentation (MEDA) framework, covering both non-incremental and incremental learning settings. The experimental results of both public and business datasets show that MEDA effectively achieves the desired effect of data augmentation and MEL can outperform the conventional SEL by a significant margin. Furthermore, MEDA's deployment in a real-world online advertising system and subsequent A/B testing demonstrate its substantial benefits in practical applications.

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

Table 5: The test loss performance on the public datasets. MEDA methods run 2 epochs. The smaller the better.

(a) Amazon

|  | DNN | DIN | DIEN | MIMN | ADFM |
|---|---|---|---|---|---|
| Single-Epoch | 0.2512 | 0.2403 | 0.2373 | 0.2241 | 0.2436 |
| MEDA-NI | **0.2423** | **0.2322** | **0.2320** | **0.2125** | **0.2387** |
| Improv. | -0.89% | -0.81% | -0.53% | -1.16% | -0.49% |
| MEDA-I | **0.2418** | **0.2327** | **0.2321** | **0.2137** | **0.2388** |
| Improv. | -0.94% | -0.76% | -0.52% | -1.04% | -0.48% |

(b) Taobao

|  | DNN | DIN | DIEN | MIMN | ADFM |
|---|---|---|---|---|---|
| Single-Epoch | 0.2255 | 0.2196 | 0.1946 | 0.1583 | 0.1424 |
| MEDA-NI | **0.1965** | **0.1708** | **0.1707** | **0.1465** | **0.132** |
| Improv. | -2.90% | -4.88% | -2.39% | -1.18% | -1.04% |
| MEDA-I | **0.1942** | **0.1653** | **0.1704** | **0.1435** | **0.1322** |
| Improv. | -3.13% | -5.43% | -2.42% | -1.48% | -1.02% |

Table 6: The test loss performance on the business datasets. MEDA methods run 2 epochs. The smaller the better.

|  | Short-Video Order | Short-Video Search LTV |
|---|---|---|
| Single-Epoch | 0.06628 | 0.14945 |
| MEDA-NI | **0.06501** | **0.14713** |
| Improv. | -0.13% | -0.23% |
| MEDA-I | **0.06512** | **0.14748** |
| Improv. | -0.12% | -0.20% |

# A    APPENDIX

## A.1    ADDITIONAL IMPLEMENTATION DETAILS

For public datasets, for DNN, DIN, DIEN, MIMN, and ADFM, the MLP has 3 layers with widths of [200,80,2]. For DIN, DIEN, MIMN, and ADFM, an additional attention module is an MLP of 3 layers with widths of [80,40,2]. All the other configurations and hyper-parameters are set according to the optimal hyperparameters reported in their respective papers without modification.

For business datasets, for the Short-Video Order dataset, the MLP has 4 layers with widths of [1024, 256, 128, 2], while for the Short-Video Search LTV dataset, the MLP has 4 layers with widths of [1024, 512, 256,1].

## A.2    THE RESULTS OF LOSSES

We provide test losses for our main results in the Tables 5 and 6. The results are consistent with those in Tables 1 and 2 of the main paper.

## A.3    ABLATION STUDY

Due to the space limit, some ablation studies are conducted on the Amazon dataset only, for whose overfitting issue is more severe.

***Hyperparameter Robustness***. We conducted additional ablation studies for various initializations and hyper-parameters for DNN on the Amazon dataset. We tested six initializations: Glorot, initializing with all ones (Ones), a uniform initializer with the range of 0.01, and normal initializers with zero mean and standard deviation of 0.01, 0.1, and 1. The results are shown in Figure 9. The figure shows that our default initializers, i.e., Glorot and a uniform initializer with the range of 0.01, have similar results with a normal initializer with a standard deviation of 0.01. For normal initializers with a standard deviation 0.01, the larger the standard deviation, the worse the entire results, maybe due to larger initial biases that are difficult for the model to learn. Initializing with all ones has the worst results, maybe because it renders IDs difficult to differentiate between each other. Therefore, based on these results, the initial biases between IDs cannot be too large or too small. Taking the default value of standard deviation or range of single-epoch learning is OK.

In Figure 9, we examine the impact of reversing the embedding order or omitting some (even or odd) groups of embeddings in training $\mathcal{D}_{tr}^2$ using our incremental MEDA. Both types of variants result in compromised performance, indicating that either training on old embeddings or reducing data augmentation is inferior. Nonetheless, these variants still demonstrate enhanced performance and robustness of our MEDA methodology across increasing epochs without overfitting.

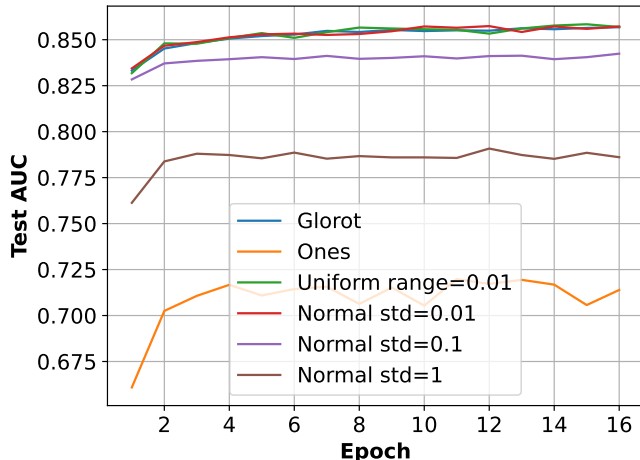

Figure 8: The test AUC curves with DNN on the Amazon dataset, comparing different variants of our incremental MEDA.

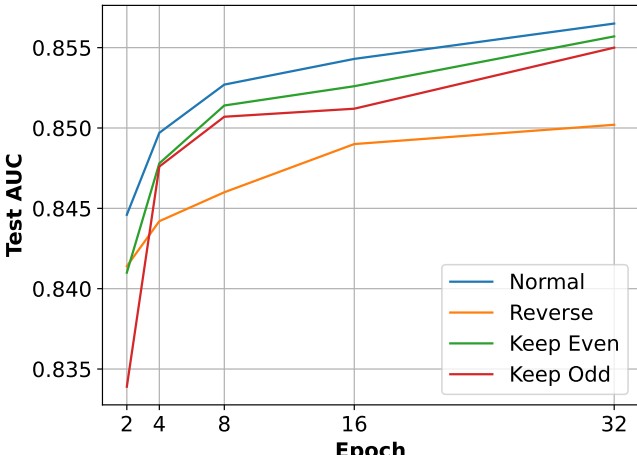

Figure 9: The test AUC curves with DNN on the Amazon dataset, comparing different variants of our incremental MEDA.

### A.4 THEORETICAL ANALYSES

The OEO results from the embedding layer excessively memorizing each single data sample, where the overfitting starts exactly at the onset of the second epoch, when the embedding layer trains on the exactly same data samples that it has trained on in the first epoch. Therefore, the key to solving the excessive memorization problem is to control memorization. We provide a theoretical foundation of differential privacy (Chaudhuri et al., 2011; Dwork et al., 2014) for the memorization control. Because differential privacy can theoretically control the amount of information of any input data sample obtained from the learned parameters.

Here we discuss this in detail. Let $\tilde{E}_1$ be the learned embedding parameters at the end of the first epoch. Then let $E_2 = \tilde{E}_1 + b$ be the initial embedding parameters at the beginning of the second epoch, where $b$ is the noise matrix with the same size of $\tilde{E}_1$. And assume that the loss funcion of the first epoch is added with the $\ell_2$ normalization of the embedding with the coefficient of $\Lambda$, and each $i$th row vector of $b$, $b_i$, is a random noise vector with the density of $v(b_i) = \frac{1}{\alpha_i} \exp(-\frac{n_i \Lambda \epsilon}{2} ||b_i||)$, where $\alpha_i$ is a normalizing constant, $n_i$ is the number of samples corresponding to the $i$th embedding vector. Then under some mild condition, the Theorem 6 of Chaudhuri et al [1] guarantees that

such a noise addition algorithm provides $\epsilon$-differential privacy for each $i$th row of $E_2$, such that for each $i$, for any two data sets $\mathcal{D}$ and $\mathcal{D}'$ that differ in a single sample and for any set $\mathcal{S}$, we have $\exp(-\epsilon)\mathbb{P}(E_2^i \in \mathcal{S}|\mathcal{D}') \leq \mathbb{P}(E_2^i \in \mathcal{S}|\mathcal{D}) \leq \exp(\epsilon)\mathbb{P}(E_2^i \in \mathcal{S}|\mathcal{D}')$, where $E_2^i$ is the $i$th row of $E_2$. The theoretical result means that, for anyone, it is difficult to know whether the input data set has changed any single data sample from $E_2^i$. In other words, $E_2^i$ contains little information about any specific data sample in the first epoch. The amount of information is precisely controlled by the pre-set hyper-parameter of $\epsilon$. Based on the above theoretical foundation, our methods of both non-incremental MEDA and incremental MEDA can be regarded as letting $E_2 = \tilde{E}_1 \times 0 + b$, which for differential privacy equals letting $E_2 = \tilde{E}_1 + b \times (+\infty)$, then it can be achieved by setting $\epsilon = 0$, then we can guarantee the perfect 0-differential privacy, such that for each $i$, for any two data sets $\mathcal{D}$ and $\mathcal{D}'$ that differ in a single sample and for any set $\mathcal{S}$, we have $\exp(-0)\mathbb{P}(E_2^i \in \mathcal{S}|\mathcal{D}') \leq \mathbb{P}(E_2^i \in \mathcal{S}|\mathcal{D}) \leq \exp(0)\mathbb{P}(E_2^i \in \mathcal{S}|\mathcal{D}')$, where $E_2^i$ is the $i$th row of $E_2$, which means that $\mathbb{P}(E_2^i \in \mathcal{S}|\mathcal{D}') = \mathbb{P}(E_2^i \in \mathcal{S}|\mathcal{D})$, suggesting that $E_2^i$ contains no information of any specific data sample in the first epoch, then the excessive memorization is perfectly solved.

On the other hand, the data augmentation property is another reason that our method can improve prediction performances. One common type of data augmentation is changing input data samples without changing the corresponding labels. Our method belongs to this type. Specifically, through different initializations, our method changes the embedding space, i.e., the linear projection matrix, of categorical features, without changing the CTR labels and the categorical values (i.e., IDs). Therefore, for the MLP layers, the input data samples mapped by the linear projection matrix are changed, but the corresponding CTR labels are kept the same. Therefore, we call our method a data augmentation method. Furthermore, the above type of data augmentation usually requires that the semantic meaning of the input data sample will not be changed. Such property is also assumed to be held in our method, because the CTR labels and the categorical values (i.e., IDs) are kept the same. Therefore, the similarity relationships between IDs — the semantic meaning of the input data sample — are still kept. The theoretical foundation of the data augmentation ability can be referenced to dropout (Srivastava et al., 2014), and the reason to improve prediction performances is also model ensemble.

