# OpenReview forum: "Multi-Epoch Learning with Data Augmentation for Deep Click-Through Rate Prediction"
_ICLR.cc/2025/Conference — ICLR 2025 Conference Withdrawn Submission_

### Official Review · Reviewer_Ci45 · 2024-10-27

**Soundness:** 3
**Presentation:** 1
**Contribution:** 2
**Rating:** 3
**Confidence:** 4

**Summary:**

This paper studies the multi-epochs overfitting problem in CTR prediction tasks. A solution that maintains multiple groups of embeddings is introduced to solve the problem. Experiments on five datasets are conducted to demonstrate the effectiveness.

**Strengths:**

1. Provide an effective solution that addresses the multi-epochs overfitting problem.

2. Extensive experiments on public datasets and industrial scenarios across several baselines.

**Weaknesses:**

1. The key insight in Figure 2 needs more analysis. "Data 1 Emb as Fixed" also leads to overfitting, which may be caused by the new categorical ids in Data 2. Please conduct an analysis separating the performance between already-seen IDs and new IDs.

2. Limited contribution. The reason for multi-epoches overfitting has already been studied in [1]. The proposed solution, especially for incremental learning, seems just to maintain k-replications (with different but random initialization) embeddings for k epochs respectively, to avoid learning from already fitted embeddings. I suggest that the authors should provide a more in-depth analysis of how the provided solution differs from or improves upon simply maintaining k-replications of embeddings.

3. Writing issue: some concepts are not well-explained. Does the absolution position (in Sec 4.1) refer to the feature sign? What does the training algorithm (in Sec 4 Non-incremental Learning) mean? Why do multiple groups of embedding form distinct embedding spaces?

4. Reproducibility and Open-Sourcing: Details in the experiments are not clear, and the paper does not mention if the code will be made available public. Ensuring reproducibility is critical for broader validation and adoption of the proposed techniques.

**Questions:**

See above.

---

### Official Review · Reviewer_8P5f · 2024-10-31

**Soundness:** 4
**Presentation:** 4
**Contribution:** 4
**Rating:** 6
**Confidence:** 4

**Summary:**

The authors propose a novel Multi-Epoch learning with Data Augmentation (MEDA) framework, covering both non-incremental and incremental learning settings. The experimental results of both public and business datasets show that MEDA effectively achieves the desired effect of data augmentation and MEL can outperform the conventional SEL by a significant margin.

**Strengths:**

1. The one-epoch problem has long been an unresolved challenge, and I’m glad to see the authors making further in-depth exploration in this area—it’s truly fascinating.
2. According to my understanding, the core of Non-incremental MEDA is that the MLP part is trained normally, and the Embedding part is randomly initialized with each epoch. Simple approach, experimentally effective.

**Weaknesses:**

1. The authors claim that it is easy to overfit the Embedding layer at the first epoch, so why does the collaborative filtering model not have this problem?
2. Algorithms 1 and 2 are not written clearly. For example, the $E_r$ for lines 276 and 277 look the same, but why do I have to repeat the explanation. Feel that the writing is relatively messy, and can not make people understand what the algorithm is doing at once.
3. In Figure 3, the AUC without MEDA, why is the training phase flat, but the test phase shows a zigzag shape?
4. In Figure 3, the AUC rises zigzag in the presence of MEDA. In Figure 5, why is it smooth again? Their horizontal and vertical coordinates are the same.

**Questions:**

See weakness

---

### Official Review · Reviewer_JMbv · 2024-11-01

**Soundness:** 3
**Presentation:** 3
**Contribution:** 2
**Rating:** 3
**Confidence:** 4

**Summary:**

This paper investigates the one-epoch overfitting (OEO) phenomenon in CTR prediction models and empirically identifies the overfitting of embedding parameters as the primary cause. Based on this insight, the authors propose a method of randomly reinitializing embedding parameters at the beginning of each epoch and continually learning the parameters of MLPs. This proposed method is simple yet effective, enabling multi-epoch learning.

**Strengths:**

The one-epoch overfitting (OEO) phenomenon is a common issue often encountered in industry and has been reported in previous work. This paper specifically investigates this issue and proposes a simple workaround to enable multi-epoch learning, thereby improving model performance. In summary, the paper has the following strengths:
+ The paper reproduces the OEO phenomenon in open benchmark datasets, validating its existence.
+ Extensive experiments and analyses are conducted to identify the main cause of OEO, which is the overfitting of embedding parameters.
+ The paper presents a simple solution to avoid OEO by randomly reinitializing embedding parameters at the beginning of each epoch, a method tailored for both incremental and non-incremental learning scenarios.
+ Both offline experiments and online A/B tests have been conducted to validate the effectiveness of the proposed method.

**Weaknesses:**

I think the paper has the following major limitations that need further improvement.

1. The paper claims, 'We provide theoretical analyses of the reason,' but I did not find any theoretical analysis throughout the paper. The authors are suggested to give a theoretical analysis before the approach.
2. The authors primarily experiment with the Amazon dataset and presume the existence of one-epoch overfitting (OEO). However, according to the results reported in the benchmark paper 'Open Benchmarking for Click-Through Rate Prediction,' OEO does not always occur (some datasets exhibit OEO, while others do not; some models experience OEO, while others do not). Given these variations, the paper lacks a rigorous analysis of the conditions under which OEO occurs. Comparing datasets where OEO occurs with those where it does not could better help identify the causes, for example, by highlighting differences between the datasets.
3. While the proposed method to avoid OEO is simple, it introduces multiple groups of embedding parameters in the incremental MEDA setting. Given that the size of embedding parameters is substantial, this approach will require multiple times the memory resources. I am unsure whether the return on investment (ROI) is sufficiently high to justify this increased resource usage. The authors are suggested to include some metrics on memory usage to indicate the practicality of the method.
4. The method is simple and effective. But the authors lack a discussion whether some alternatives are considered, for example whether using a smaller learning rate for embeddings and a larger learning rate for MLPs could mitigate the OEO issue.

**Questions:**

1. The term SEL is not defined upon its first occurrence.
2. In the related work, the authors mention graph learning and state that "graph learning research has delved into fine-tuning pre-trained models for new graphs." However, there is no clear connection drawn between graph learning and CTR prediction research, particularly for this paper. The authors are suggested to explicitly state the relevance of graph learning to the work on CTR prediction, or just remove this section if it's not directly related.
3. The paper states, "Figure 3 shows that the embedding overfits on each trained data sample," but the plot does not track individual data samples. It is unclear how the authors determine that overfitting occurs on each sample. For example, the training loss may drop and then slightly increase without MEDA, and the testing loss may also fluctuate. The authors can clarify how they drew this conclusion from the data presented in Figure 3.
4. The paper claims, "the initial embedding of the second epoch precisely memorizes the information of any data sample in the first epoch." However, there is no empirical evidence provided to support this statement. Given that models typically experience information loss during training, this claim seems unlikely. The authors are suggested to provide empirical evidence supporting this claim, or to clarify if this is a hypothesis rather than a proven fact.
5. The sentence "Because, first, Figure 2(a)..." contains a syntax error.
6. On line 269, the figure reference may be incorrect. The text states, "Furthermore, our results in Figure 4(a)..." but the figure shows the cosine similarity between MLPs, not the convergence of MLPs.
7. On line 311, the details about incremental MEDA are unclear. Specifically, the algorithm "selected based on requirements such as computation/storage costs" is not described. How are these selections made according to computation and storage costs? Additionally, Figure 1(b) mentions two operations, "copy" and "select," but the "copy" operation is not described in Algorithm 2.
8. In the A/B testing section, the authors indicate that they use incremental MEDA. However, the reported numbers (test AUC increase, duration of A/B test, training time) are identical to those reported in the first Arxiv version, which only discussed non-incremental MEDA in that experiment. It is unclear whether two separate A/B tests were conducted or if the same results are being reused.

---

### Official Review · Reviewer_1BL9 · 2024-11-01

**Soundness:** 3
**Presentation:** 3
**Contribution:** 2
**Rating:** 3
**Confidence:** 5

**Summary:**

This paper targets the single-epoch phenomenon in the CTR domain. The author identifies the overfitting as coming from the embedding layer. Specifically, a multi-epoch data augmentation method(MEDA) is proposed to study the disentangling of the dependency between embedding and the MLP layer. Both an incremental and non-incremental approach are proposed. The proposed method has proven effective with both online and offline experiments.

**Strengths:**

1. The proposed MEDA framework is simple yet effective on the listed datasets.
2. The proposed method has proven effective with both online and offline experiments.
3. The presentation of this paper is easy-to-follow.

**Weaknesses:**

1. The hyper-parameter in all experiments is not carefully tuned. Hence, we don't know if the effectiveness of MEDA is caused by the method itself or the weak baselines.
2. The generalizability of the one-epoch phenomenon and MEDA is over-emphasized. Not all CTR datasets witness a one-epoch phenomenon. For instance, no paper ever reported such a phenomenon on Criteo, which can be considered one of the most important datasets in CTR prediction. Based on the personal experimental experience, the author personally feels Criteo does not fit into this case.
3. The paper also lacks a theoretical analysis of what causes the one-epoch phenomenon and why MEDA can alleviate such a phenomenon. Appendix A.4 fails to make the reviewer understand these points.
4. MEDA is only tested on the MLP backbone. Other backbones such as transformer and resnet are missing in this paper.
5. The paper lacks discussion and comparison with existing model retraining techniques.
6. No code artifact is provided, undermining the reproducibility of this paper.

**Questions:**

See weaknesses

---

### Official Review · Reviewer_tXTT · 2024-11-03

**Soundness:** 2
**Presentation:** 3
**Contribution:** 2
**Rating:** 3
**Confidence:** 4

**Summary:**

This paper investigates OEO, considering overfitting in the embedding layer caused by high-dimensional data sparsity as a critical issue. To address this, it proposes a framework called Multi-Epoch learning with Data Augmentation (MEDA), which decouples the embedding layer and the data and reinitializes embedding parameters at each epoch. The results demonstrate its effectiveness in non-incremental and incremental learning scenarios, with benefits observed in real-world applications.

**Strengths:**

1,Mitigates OEO Phenomenon: By reinitializing embedding parameters at each training epoch, the proposed approach (MEDA) effectively mitigates the Over-fitting Embedding Overhaul (OEO) phenomenon, leading to enhanced performance in CTR prediction models.

2,Thorough Analysis of Problem Causes: The paper provides a detailed analysis of the underlying causes of the embedding overfitting issue, showing a strong grasp of the challenges inherent to CTR prediction.

3,Promising Experimental Results: Results across multiple datasets demonstrate the proposed method’s effectiveness, adding empirical support to its benefits.

4,Clarity of Presentation: The clear and well-organized writing style makes the paper easy to follow, especially for readers unfamiliar with the nuances of CTR modeling.

**Weaknesses:**

1,Limited Novelty of Proposed Framework: The innovation in this work appears limited to reinitializing embeddings each epoch. The paper suggests that embedding overfitting stems from initial embeddings over-memorizing data specifics across epochs; however, this can often be mitigated by reshuffling the data between epochs, an approach that isn’t fully explored here.

2,Lack of Experimental Breadth: The experimental comparisons mainly focus on one-epoch versus multi-epoch MEDA trials across different CTR models (e.g., DIN, DIEN), but they do not include other techniques for addressing embedding overfitting, such as embedding dropout, data reshuffling, or regularization. Including these would provide a more comprehensive evaluation of MEDA’s effectiveness.

3,Marginal AUC Improvement in Online Experiments: While the online experiments indicate a slight AUC improvement, there’s limited evidence of improvement in CTR, the paper’s primary focus. Since CTR is critical to the study, clearer gains in CTR would make a stronger case for MEDA, alongside secondary metrics like revenue.

4,Training Instability Due to Frequent Reinitialization: Reinitializing embeddings every epoch may cause instability, as the MLP layer must continually adapt to the changing embeddings, leading to performance fluctuations. A comparison with alternative methods like data reshuffling could offer a clearer view of MEDA’s trade-offs, particularly around stability.

**Questions:**

please check the weaknesses.

---

### Note · Authors · 2025-01-16

I have read and agree with the venue's withdrawal policy on behalf of myself and my co-authors.